# Application of Pattern Language for Game Design in Pedagogy and Design Practice

**Christopher Aaron Barney**

GAME and GSND Departments, College of Arts Media and Design, Northeastern University, Boston, MA 02115, USA; c.barney@northeastern.edu; Tel.: +1-(671)-939-7868

**Abstract:** Existing implementations of game design patterns have largely been confined to theoretical or research settings. Weaknesses in these implementations have prevented game design patterns from being properly evaluated as an educational and practical development tool. This paper examines these weaknesses, describes a method of developing and applying patterns that overcome the weaknesses, and evaluates use of the method for game design education and practice. Weaknesses in existing pattern implementations are: the omission of design problems, presumption of functional completeness at the level of pattern languages, narrow topical focus, and lack of a concise, repeatable method for pattern production. Several features of the proposed method were specifically built to address these weaknesses, namely the pattern template, the process for connecting patterns into a language and assessing the language's scope, a rubric for assessing pattern confidence and interconnectivity confidence, and pattern-building exercises. This method was applied in a classroom setting. Results as assessed by the evaluation of student work suggest that creating patterns/pattern languages is an effective pedagogical approach. Designs produced using designer-created patterns closely align with existing design theory and are clearly understood by students. The above results may indicate that the path to gaining wider acceptance of pattern theory as a design framework within game design is not to produce a universal pattern language, but to facilitate the creation of case-specific languages by students and professional designers that use a shared ontology, and thus can be combined easily to solve the diverse sets of problems faced by these groups.

**Keywords:** game design; design patterns; pattern language; design pattern application; design pattern creation

## 1. Introduction

Game design as an industry operates under a significant number of constraints, from financial to labor to audience. Seeking optimal solutions, it operates in a constant state of self-definition. Game design as an academic discipline is likewise in a state of rapid change as it tries to balance a pursuit of understanding of the discipline as manifested in the products of the industry, and the more measured development of theory and practice based on the research interests of scholars.

The problem of how to study, teach, and practice game design has proven to be non-trivial. Practically these three goals have been pursued independently with varying degrees of success. Profound games such as Train [1] or This War of Mine [2] have been made, as well as financially successful ones such as Call of Duty [3] or Fortnite [4]; occasionally, these two coincide, as in the case of Spec Ops: the Line [5]. Insightful studies are conducted regularly, and game scholars leverage their design insights to produce serious games for educational, research, and training purposes. Games Studies programs such as those at Full Sail University and DigiPen Institute of Technology have been developed to help train students as practical developers, and others, such as the Game Science and Design program at Northeastern University, have been created to train students as games

researchers. Broadly, though, these three areas of development have failed to find a shared ontology that allows advances in one to be readily applied to the others. Frameworks such as Mechanics Dynamics and Aesthetics (MDA) [6], Four Keys to Fun [7], The Five Domains of Play [8], or Jessie Schell's Design Lenses [9] have proven very useful. However, none of these specific attempts at creating a framework addresses the full scope of game design [10,11]. Each has been met with resistance by some of the proponents of the others.

Researchers, educators, and game designers have failed to agree on a shared ontology because they have divergent needs. Even within the game development industry, the needs of one set of developers differ radically from those of another to the degree that having a shared design vocabulary can seem unproductive.

This paper argues that the solution is not to produce a universal framework, but to develop case-specific frameworks that use a shared ontology and thus can be combined easily to solve the diverse sets of problems faced by these groups. The solution can be implemented by applying the idea of design patterns and pattern languages to the game design space.

The work described in this paper differs from and extends the work published in the book Pattern Language for Game Design [12] in that it proposes a formal ontology for the pattern template and describes the ways that the pattern creation process maps to the learning and teaching functions described by Shuell and Moran [13]. Additionally, this paper describes the application of the pattern creation process in both an educational setting and as part of practical development in an industry setting. This establishes methods for applying a clear pattern language creation process that, through iterative development, has reached a stable state where formal validation can begin. The validation process is described in the Section 4 of this paper is ongoing. Results will be presented in future publications as they become available.

### 1.1. What Is a Design Pattern?

The concept of a design pattern was introduced in 1977 by Christopher Alexander as a solution to problems he saw in architectural design. He defines an architectural pattern as

> "a careful description of a perennial solution to a recurring problem within a building context, describing one of the configurations that brings life to a building. Each pattern describes a problem that occurs over and over again in our environment, and then describes the core solution to that problem, in such a way that you can use the solution a million times over, without ever doing it the same way twice." [14]

The use of patterns has developed across disciplines over the intervening forty years and will be discussed in the next section. The core definition of a pattern has been consistent throughout this period of development, though different implementations have emphasized or minimized aspects of Alexander's initial concept.

### 1.2. A Brief History of Pattern Development and Use

Alexander's book A Pattern Language: Towns, Buildings, Construction articulated his rationale for creating design patterns related to architecture. This book detailed 253 patterns and linked them together into a loose hierarchical language. The specifics of this work as it relates to game design patterns are discussed in more detail in Pattern Language for Game Design [12].

Sometime after the publication of A Pattern Language, Alexander's work attracted the attention of computer scientists. By the 1990s, a group of four scholars had adapted his ideas for use in object-oriented programming in the book Design Patterns: Elements of Reusable Object-Oriented Software [15]. This book defined 23 patterns for use in this domain. The patterns produced by this group clarified thinking around object-oriented programming and became a central organizing force in the practice in the decades since

their publication. This effect demonstrates the utility of pattern theory. However, this limited set of patterns was seen as sufficient, and likely compete. Additionally, these patterns focused on the details of each solution and largely omitted the specific design problems that the patterns solved. Lastly, the focus of this small language was on object-oriented programming, which is in and of itself reasonable. However, many programmers saw the utility of these patterns and used it as an argument for the exclusiveness of the object-oriented paradigm. These three shortcomings in the language, rather than in the validity of the patterns, have occurred to a greater or lesser degree in many of the pattern's subsequent language applications.

In the four decades since the publication of Alexander's originating work, patterns have spread to a number of other fields, the most notable being social and behavioral science. In particular, the project by Takashi Iba to develop a pattern language for creative learners describes his large-scale project to develop a pattern language [16]. This project is notable in that while it does propose a fixed language, it also clearly describes a repeatable process that was followed to create the language.

### 1.3. Pattern Projects in Games

Existing general pattern languages show some of the methodological shortcomings discussed in the previous section: presumed completeness, omission of design problems, or undeclared narrowness of focus. Examples are the two seminal pattern languages in games created by Adams and Dormans and by Björk and Holopainen, respectively.

In the book Game Mechanics: Advanced Game Design [17], the authors created a small pattern library that was narrowly focused on game systems mechanics. The versatility and atomic nature of their patterns was very impressive. However, the narrow mechanics-focused scope of their patterns limits the ability to directly connect patterns from this library with those from different areas of design or to understand how patterns from these different areas might interact with those from this library. It also introduces the risk of privileging those aspects of mechanical design over other parts of the discipline.

The scholars Stafan Björk and Jussi Holopainen first published a paper titled Patterns in Game Design [18] and later a book titled Game Design Patterns [19] where they began to produce a large pattern library. This library shifts focus heavily to the pattern and away from the problem that the pattern addresses. The library of patterns produced by this work has grown to include more than 700 patterns [20]. Unlike previously discussed languages, it is generalized and covers a broad section of the field. It is also not considered complete and continues to grow over time. However, the breadth and depth of this repository can limit its usefulness to scholars and developers who have not been part of its production. The scale of the library exacerbates the problem created by omitting design problems, and results in difficulty understanding what patterns are applicable under what circumstances. Ongoing work on this project is beginning to address this problem using visualization techniques to improve discoverability [21].

There are a significant number of other game design-related projects, and I am limiting my discussion here to these two as they have had the broadest impact. However, the vast majority of existing projects have at least one of the above faults: narrow focus, presumed completeness, or failing to state the design problem addressed by each pattern.

### 1.4. Using Patterns in Pedagogy

An interesting use of patterns as part of creating a pedagogy can be seen in a set of papers by researchers of serious games for education. The first of these is Design Patterns for Learning Games [22] which proposes combining the use of game design patterns drawn from the library created by Björk and Holopainen, pedagogical patterns from the Pedagogical Patterns Project [23], and the twenty-two learning and teaching functions proposed by Shuell and Moran [24] to create games that effectively teach a particular knowledge set. These ideas are further developed and tested in the papers Game Design Techniques for Software Engineering Management Education [24] and Teaching Software

Engineering Topics Through Pedagogical Game Design Patterns: An Empirical Study [25]. The results of the studies indicate that the use of patterns is effective in designing games that effectively teach specific subject matter. This work differs from this project in that the patterns being used are static and drawn from an external source, and that the source of learning is the produced game artifact, whereas the process discussed in this paper uses the development of patterns to teach the discipline of game design as well as the use of patterns to produce games with intended effects.

The process for creating and using patterns described in this paper was created independently from the above line of research. However, the Section 3 of this paper provides an assessment of the process I introduce here in light of the twenty-two learning and teaching functions described by Shuell and Moran and creates a mapping to the process used for pattern creation discussed in this paper. This full mapping is included as Supplementary Materials.

### 1.5. An Alternative Application of Patterns

The application of patterns as described in this paper arose out of a practice-driven application of pattern theory in the classroom and as applied to small-scale development projects. In the classroom, the process described in this paper was developed and used over consecutive six semesters in four different courses. Specifically, the process was introduced in the graduate course Spatial and Temporal Design, where it has been used three times. It was then used for the undergraduate course Level Design and Game Architecture, where again it has been used three times, and it has most recently been added to the undergraduate course Foundations of Game Design where it has been used once. Additionally, the process has been used by the developer Christopher Totten to develop the level design paradigm for his studio's current project, and by the Google and ArenaNet developer Lincolin Hughes to investigate the foundations of social design.

These applications of the process were undertaken with several goals. First, to use the process of deriving a pattern to teach a specific design principle. Second, to allow a cohort of developers to create a shared design vocabulary that both serves their specific needs and is understood by all participating developers. Third, to help developers create an individual conceptual design framework that fits their learning style and development needs rather than providing rote learning of existing frameworks such as Mechanics Dynamics Aesthetics (MDA) [6] or Design Lenses [9].

To achieve these goals, it was necessary to create and codify the process of pattern creation. In general, the process of creating a pattern has been omitted in the literature, even within Alexander's work. Takashi Iba discussed his methods in some detail, but his process was designed for a large group able to dedicate significant resources to the development of a language [16]. For active use in a classroom and in small-scale development, a more concise scalable approach would be needed.

### 1.6. Proposal for Individual Patterns and Languages

The project undertaken to meet these goals was to develop a repeatable process for creating patterns that could be executed by designers of any skill level, and which would produce patterns that could be clearly understood and used by any designer familiar with the process. Further, this process of language creation allows designers to link the patterns they created, as well as existing patterns, together to form case-specific pattern languages that can produce reliable solutions to complex design problems. Finally, the process allows individual developers to organize their personal design knowledge as best fits their needs. Thus, while individual patterns and small problem-driven languages are created independently, they are interoperable and can be combined to address the problem set that defines the entirety of a given design.

## 2. Materials and Methods

To satisfy these needs, I created a pattern template that could be used to make sure that all patterns would contain all of the same ontological structure. The pattern template created was based on the original ideas of Alexander as laid out in his seminal work A Pattern Language: Towns, Buildings, Construction. I incorporated ideas from the patterns of Björk & Holopainen as well as Adams and Dormans where they were in line with my understanding of Alexander's intent. Additionally, I added elements that I felt were needed to support learning through the process of pattern creation, as well as the use of the created patterns in practical design.

To ensure a uniform application of this template, as well as facilitate the use of patterns and pattern languages recorded using the template, I created a web-based tool for entering and accessing patterns. This tool uses a relational database to store pattern data. The creation of this database schema implied an ontological structure which will be discussed later in the Proposed Ontology for Pattern Language for Game Design Patterns section. Additionally, the creation of a public-facing repository for recording patterns necessitated the consideration of issues of privacy and access for patterns recorded using this tool. Aspects of the template that relate to these concerns are presented in the second optional part of the template as they are not intrinsic to patterns but are integral to this practical implementation of a pattern library.

Using this template, a series of exercises were created and implemented in a classroom setting. The resulting patterns were reviewed in terms of how well they served the goals derived from the proposal for individual patterns and languages:

- Pattern reflects or is not in conflict with validated design principles
- Pattern can be clearly understood and used by any designer
- Pattern describes its links with other existing patterns
- Pattern addresses the specific problem of the designer creating it

As stated in the section "An alternative application of patterns", these exercises were iterated on over the course of six semesters of courses. The resulting pattern derivation process and language creation process are described below.

Pattern creation and application was incorporated into three courses at Northeastern University. Each course covered a long-standing aspect of the game design curriculum in their respective programs. Some of these aspects are course-specific, such as the architectural and level design understanding in the Spatial and Temporal Game Design course and the Architecture and Level Design course. Other aspects are general and span all courses in the curriculum, for example the ability to analyze existing design and understand its purpose, or to look at specific design techniques and consider their effects across diverse audiences. At the time that this method was introduced to these courses, none of the students had encountered the method in a previous course. Going forward, the effect of students encountering the method in a foundational course and then revisiting it in advanced and graduate courses will be evaluated. The specifics of how the below method was incorporated into these courses is detailed in the Discussion of pattern exercises as part of curriculum section.

### 2.1. The Iterative Process Resulting from Common Problems

The creation of the process described in this paper has been very iterative. Patterns were first introduced as a part of course pedagogy before the format of exercises had been established. After each use of patterns in course work, the structure of exercises and the specifics of the template used to record patterns was refined. Additionally, the process of linking patterns together into a language was initially performed in a very manual way where all students reviewed each pattern in the initially small library and looked for connections. As the library grew, the techniques of using keywords, pattern seeds, and recommended exercises were added. Much of the functionality of the pattern library website was developed in support of this process.

*2.2. Revised Pattern Definition/Template*

In order for the patterns created by this process to meet the stated goals, each pattern must contain all of a set of properties and may contain additional optional properties. The required parts of a pattern are:

- Pattern Title
- Design Problem
- Pattern Description
- Pattern Confidence
- Author(s)
- Keywords (Keywords, Categories, Properties)
- Example Games (Name, Example Description)

The optional parts of a pattern are:

- Pattern State
- Pattern Seed
- Groups(s)
- Pattern Image
- Pattern Image Description
- Related Patterns (Name, Description, Confidence)
- Suggested Exercises

2.2.1. Definitions of Required Properties Are Presented Here

**Pattern Title:** The name of the pattern. Alexander advised that the name of a pattern should be both descriptive and evocative. In practice creating a good title can be challenging as it is also important not to create a title that requires a niche cultural context.

**Design Problem:** The general design problem that the pattern serves as a solution to. When setting out the template and displaying patterns, it is important to privilege this property as it is common for developers to spot a pattern and document it without considering its design problem as seen in Game Design Patterns [19] and early student patterns in this project.

**Pattern Description:** This is the heart of the pattern, a description of the actual pattern. The description has a length between one paragraph and one page. It is important to give enough context to make the pattern easy to implement and to call out drawbacks or concerns about the pattern. In general, beginning with a sentence like "In order to [achieve some design effect], a designer may [take some design action, use some mechanic, etc.] because [explanation of how the pattern produces the desired effect]."

**Pattern Confidence:** This is a numerical reflection of the author's confidence in the accuracy and effectiveness of the pattern. This number is generated using a confidence rubric that will be presented below. The addition of a formal rubric for confidence is an enhancement to the idea of confidence as introduced by Alexander in A Pattern Language [14].

**Author(s):** All of the designers who contributed to the creation of the problem.

**Keywords:** A listing of all of the keywords that relate to the pattern. This is critical in linking patterns together when they do not have a more structured connection listed in the property of the related pattern. Keywords are also important as they can be used at the time of pattern creation to search the library for other patterns that may need to be listed in the property of the related pattern. In the development of this process, a curated list of keywords was used. This list was expanded each semester, but limiting the list helps to encourage a more limited set of keywords that avoids synonyms and redundancies such as using 'Third Person' and '3rd Person.'

**Example Games:** Because, as Alexander defined it, the nature of a pattern is to be the "core solution to that problem, [stated] in such a way that you can use the solution a million times over, without ever doing it the same way twice" [14], it is necessary to include examples of the pattern's use in a diverse set of games. The process of derivation which is

discussed below, generates this list of games and capturing it here allows other designers to take the generalized pattern which can be abstract and see it applied in concrete ways.

2.2.2. Definitions of the Optional Properties Are Presented Here

**Pattern Image:** An iconic image to represent each pattern. This image can help to convey the essence of the pattern and to serve as a mnemonic anchor for remembering it. Designers with animation talent have had good luck in creating animations to demonstrate the application of their patterns.

**Pattern Image Description:** A short written description of the image is both clarifying and important for accessibility in patterns intended for public use.

**Related Patterns** *(Name, Description, Confidence)***:** This property is optional only because there may not be any related patterns, especially in new libraries with a small number of preexisting patterns. In a mature library, most, if not all, new patterns will be connected to one or more other patterns as parent, child, additive, subtractive, or alternate patterns. These structured connections are the primary axioms that structure patterns into a language. All of these relationships are recorded using the following format:

**Related Pattern Name** *(Confidence: n)*: Description of how the relationship functions.

Different pattern projects have used different terminology for these connections or used slight variations on these concepts, so I will describe each relationship axiom below.

*Parent Patterns:* A pattern or several patterns that are needed by this pattern for it to function well. Typically, this follows the form; Bjork and Halopanien use the term 'Is Instantiated By.' This is accurate and descriptive, but I have chosen a less formal and possibly less accurate language to appeal to a broader audience.

*Child Patterns:* Patterns that are suggested by this pattern or require it to function well. Again, Björk and Holopainen use the term 'Instantiates.'

*Additive Patterns:* A pattern is considered to be additive when its presence makes another pattern more effective, but when each can function without the other.

*Subtractive Patterns:* A subtractive pattern makes another pattern less effective. Björk and Holopainen group additive and subtractive patterns together under a 'Is Modulated By' axiom, which does make clear that you might want to include a subtractive pattern as part of balancing gameplay.

*Alternate Patterns:* Two patterns are alternates of each other when they both solve the same design problem, but in different—and possibly mutually exclusive—ways.

One axiom that is not represented in the current ontology is the idea of an arbitrary incompatible pattern, one pattern that causes another not to function rather than just modulating its effectiveness. The pattern derivation process that I present below simply does not uncover this relationship. However, the use of patterns derived from this process may well expose this type of axiom and it may be added to the ontology in the future.

**Pattern State:** This state reflects the level of completion and privacy of the pattern. This is idiosyncratic to a shared pattern library with many users. In order to foster the use of a library in as many ways as possible, this project allows patterns to be in a draft state where only the authors can view the pattern, in a review state where any groups that the authors are members of can view and comment on it, or in a published state where the pattern is visible by anyone. There may be additional states related to the maintenance and administration of a library, such as marking a pattern as deprecated or merged with another duplicate pattern.

**Pattern Seed:** This indicates the specific exercise and design element that was used to derive the pattern. Capturing this information is necessary for the evaluation of the effectiveness of each exercise in producing the intended types of patterns.

**Group(s):** This property represents an arbitrary collection of patterns. For instance, all the patterns created by a particular team of designers, or by students belonging to a specific course section, or associated with a university or studio. This property is, at the time of writing, unique to this project. This information is only important in large-scale or public pattern libraries. However, in these cases, it allows both privacy and the ability to

easily represent smaller pattern languages within a larger library of patterns. Patterns may be included in multiple groups allowing them to be shared by use-specific languages.

**Suggested Exercises:** Often during the process of creating a pattern, designers note multiple other possible patterns. It is not always practical to document all of these patterns immediately. This property is used to record these nascent patterns so that developers can pursue them at a later time.

*2.3. Proposed Ontology for Pattern Language for Game Design Patterns*

It is valuable to codify the above properties and axioms using a formal ontology language. The OWL 2 ontology [26] was chosen for this purpose. While only a simple definitional ontology has so far been created, the long-term goal is to apply a robust ontological structure to all patterns created in a programmatic way and to instrument the pattern library, allowing existing reasoner software to facilitate the selection of pattern languages that address complex sets of design problems. This application of the proposed ontology will be the subject of ongoing research.

Because the Pattern Library website uses a MySQL RDB to store all data related to the patterns entered there, an ontology can be programmatically applied by creating declarations for all of the classes and object properties in the ontology and then using the existing PHP/JavaScript code to produce individuals for all of the patterns in the library, or in a specific language stored in the library.

A simplified representation of the ontology is presented here in Tables 1–3. The current set of classes and object properties being used are presented in formal OWL2 Manchester syntax in the Supplementary Materials.

**Table 1.** Ontology classes for representing a pattern.

| Classes | | | |
| --- | --- | --- | --- |
| example_game | game_platform | keyword | pattern_suggested_exercise |
| exercise | game_publisher | pattern | related_pattern |
| game | game_release | pattern_exercise | user |
| game_available_link | game_type | pattern_related_pattern | EquivalentTo: author |
| game_developer | group | pattern_seed | author |
| game_info_link | group_type | pattern_states | EquivalentTo: user |

**Table 2.** Ontology object properties for representing a pattern.

| Object Property - Domain - Range | Object Property - Domain - Range |
| --- | --- |
| hasAuthor - pattern - user | hasGroupType - group - group_type |
| hasExampleGame - pattern - example_game | hasKeyword - game, pattern - keyword |
| hasExercis - pattern - pattern_exercise | hasPatternSeed - pattern - pattern_seed |
| hasGame - example_game - game | hasPatternState - pattern - pattern_states |
| hasGameAvailableLink - game - game_available_link | hasRelatedPattern - pattern - related_pattern |
| hasGameDeveloper - game - game_developer | hasRelease - game - game_release |
| hasGameInfoLink - game - game_info_link | hasSuggestedExercise - pattern - pattern_suggested_exercise |
| hasGamePlarform - game - game_platform | hasUser - pattern - user |
| | owl:topObjectProperty - related_pattern - pattern |
| | relatesTo - related_pattern - pattern |

hasGamePublisher - game - : game_pub-
lisher
hasGameType - game - game_type
hasGroup - pattern, user - group

**Table 3.** Ontology data properties for representing a pattern.

| Data Property - Domain | Data Property - Domain |
| --- | --- |
| game_available_link_notes - game_available_link | pattern_exercise_name - pattern_exercise |
| game_available_link_source - game | pattern_exercise_page - pattern_exercise |
| game_available_link_url - game | pattern_image - pattern |
| game_description - game | pattern_image_description - pattern |
| game_developer_name - game_developer | pattern_name - pattern |
| game_developer_notes - game_developer | pattern_seed_description - pattern |
| game_image - game | pattern_seed_name - pattern |
| game_info_link_notes - game_info_link | pattern_state_name - pattern |
| game_info_link_source - game_info_link | pattern_suggested_execise_description - pattern_suggested_exercise |
| game_info_link_url - game | related_pattern_confidence - related_pattern |
| game_name - game | related_pattern_description - related_pattern |
| game_plarform_name - game_platform | related_pattern_type - related_pattern |
| game_platfrom_notes - game_platform | video_gameplay - game |
| game_publisher_name - game_publisher | video_trailer - game |
| game_publisher_notes - game_publisher | group_type_description - group_type |
| game_release_date - game | group_type_name - group_type |
| game_release_name - game_release | pattern_confidence - pattern |
| game_release_notes - game_release | pattern_created_date - pattern |
| game_release_type - game | pattern_description - pattern |
| game_type_name - game_type | pattern_design_problem - pattern |
| game_type_notes - game | pattern_example_game_description - example_game |
| group_auto_join - group | pattern_exercise_description - pattern_exercise |
| group_name - group | |

## 2.4. Pattern Exercises

As discussed in the introduction, a robust pattern library, while useful, is only one part of this project. The goal is to make the creation of a pattern language a repeatable process and to enable any designer to engage in the creation of patterns that fit the set of design problems they face.

Over the course of six semesters, and in three distinct courses, a variety of exercises were created to guide designers—students in this case—in creating patterns. At the time of writing, twenty-six exercises have been created and tested. The majority of these pattern exercises are presented in the book Pattern Language for Game Design [12]. The first of these will be discussed below. Students have used these exercises to create upwards of 400 patterns. In the past year, patterns have been recorded in the Pattern Language For Game Design website [27].

The Basic Pattern Exercise as presented in Listing 1presents a seven-step process for creating a pattern. It is important to note that this process does not begin with a design problem, though other exercises do. In this case, the designer is asked first to choose a design element to investigate through the exercise. For the purpose of this exercise, a design element is defined as an aspect of design that can be either intrinsic or extrinsic to the game as an artifact, and either formal or functional. For example, a platform in a side scrolling game would be an intrinsic formal element, whereas a game controller would be a formal extrinsic element. The player-controlled character's ability to jump would be an intrinsic functional element.

**Listing 1.** Statement for Basic Pattern Exercise as presented to students.

**Step 1:** Name a design element.

**Step 2:** Name ten games that use that element—the more different ways the games use it, the better.

**Step 3:** Describe how each of those games uses the element you chose. Try *not* to look for a pattern yet. Focus on accurately describing the way each game uses the element you identified.

**Step 4:** What design problems do the games use the element to solve? Some games may use the element for one purpose, while others use it for another. Many games use the elements in more than one way. Describe the problems solved by your element for each of the ten games listed in step 2.

**Step 5:** Look at steps 3 and 4. Are there patterns in the ways the games use the element, and how do those relate to the problems they solve?

**Step 6:** Pick one of those patterns and describe it using the pattern template.

**Step 7:** You may repeat step 6 for each pattern you observed.

After selecting a design element, the designer is instructed to name ten games that make use of that element. There are several considerations in this step of the process. The exercise instructs the designer to choose games that use the design element as differently as possible. This is the result of observing that patterns generated by choosing diverse games are broader, more generalizable, or higher level than those created by looking at games within a genre. This is desirable, particularly when the designer is just learning the process and when creating a new pattern language. As the designer's skills develop or if the pattern language is already robust, then more focused specific patterns might be desired.

A second concern is the number of games examined. A large sample size is good, but examining more than ten games makes the process prohibitive for a single designer. If an exercise is being completed by a group, then more than ten games should be examined. It may also be the case that a designer has not played ten games that use the design element they are focusing on. In this case, it is necessary to research additional games. Playing them is preferable but reading reviews and watching gameplay is also useful.

Once a set of games has been selected, the designer is instructed to describe how the games use the element. The designer is also cautioned against beginning the process of searching for a pattern. This was a common mistake on the part of students during the development stages of this learning method. After naming one or two games, the students would decide on a pattern and select the rest of the games in a manner that confirmed their prematurely formed pattern. The resulting patterns were usually still valid but had

a much narrower scope and often overlooked examples that would have clarified the pattern, revealed circumstances where it was less effective, or resulted in a higher-level pattern. It is not possible to eliminate this type of confirmation bias entirely, but it can be reduced by focusing on clear documentation of the use of the technique at this point.

Having produced a set of descriptions of design element use, the designer next considers what design problem each element's use solves. This process is subjective and is dependent on the insight of the designer. Though it is important to note this element of subjectivity, it does not invalidate the produced pattern; rather, it results in patterns that reflect the insights of the particular designer. The patterns produced by new designers tend to describe design principles that are already familiar to experienced designers. These simple patterns can be an important part of a larger pattern language, even when it is being shared with experienced designers. Most importantly, they allow the newer designer to deeply understand the design principle they have expressed. This has been a primary benefit to using these techniques in the classroom.

Often, several implementations of a design element address the same problem, or a single implementation addresses several disparate design problems. Thus, in the following step the designer notes all of the possible patterns that relate to their chosen element.

Finally, the designer selects one of the possible patterns and records it using the pattern template, selecting and describing appropriate example games. This last step may be repeated for each of the noted patterns.

*2.5. Connecting Patterns into a Language*

As described in the pattern template section of this paper, patterns as conceived of by Alexander do not exist independently, they exist as part of a highly connected language. The process of identifying these connections provides significant benefit for both the utility of the language and for the learning process of the designers producing the language. In the ontology used here, these connections take the form of keywords and related pattern properties.

When introducing a new pattern into an established language, it is relatively easy to make these connections using the process described below. However, in the early development of a language it may be the case that few or no related patterns yet exist. In this case, it is necessary to continue developing patterns as needed and reexamining the library with each new addition to see if new connections should also be created.

Over the course of this project, the following process has been successful for language formation.

1.  A shared set of keywords is applied to all patterns that have been created. It is important that this set of keywords is curated and limited to accurately code patterns that share concepts.
2.  All patterns that share a keyword are noted and reviewed by their authors. If either author believes that there is a relationship between two patterns, its type is discussed and agreed upon by both authors. Relationships are usually reciprocal, and thus each author needs to add the relationship to their pattern.
3.  For patterns that have suggested exercises, those exercises are completed, and the resulting patterns are linked if appropriate.
4.  If a pattern does not have any related patterns at this point, then specific exercises targeted at creating higher- or lower-level patterns are completed using the same seed as the orphan pattern. The resulting patterns are linked as indicated.

Pattern collections resulting from this process have been dense enough to constitute a useful language in of the six-semester-long iterations of this process. However, these languages have consisted of 80 to 117 patterns and are not considered to be functionally complete, or even sufficient to fully serve as a basis for pattern-driven game development. In the past four iterations of this process, all patterns were entered into the Pattern Library Website and the resulting languages were interconnected. The growing size of the Pattern

Library Website will require a process of regular trimming to remove low-quality patterns and combine patterns that are functionally identical.

*2.6. Discussion of Pattern Exercises as Part of Curriculum*

The results of the integration of pattern exercises into the curriculum have been discussed above as they apply to the refinement of the pattern and the language creation process. From the perspective of students, the generation of a repeatable process and a mature library are secondary to the role that the creation process plays in the curriculum.

In each course that incorporates patterns, students are introduced to the process of completing pattern exercises early in the course. They are then presented with different exercises depending on the content of the course. For instance, students in the introductory 'Foundations of Game Design' course complete exercises to create patterns based on formal and functional elements of design. Students in the graduate level 'Spatial and Temporal Design' course complete more complex exercises based on player circulation patterns or on boss encounter design.

In all courses, students immediately use the patterns they create in small design projects that show how they would apply the pattern to solve its stated design problem. The result of one such exercise is included in the Supplementary Materials section of this paper.

Each course progresses to the use of patterns to create a language and includes a final project in which students describe a complex design problem and then identify a small set of patterns that they believe will help them solve that problem. They then produce a midsized project over the last weeks of the course. This project is then analyzed by other students in the class to both see how well the patterns were implemented and whether they contributed to the solution of the stated design problem.

*2.7. Example of a Student-Created Pattern Language*

The following example shows a sample design problem and the pattern language used to address it. The full design document for the project is available in Supplemental Materials.

---

**Design Problem**

"Can fast and reaction based movement be made easier for newer players?"

**Patterns:**

**Architecture Boundaries**

Clear immersive boundaries make it easy to show where players can and cannot use their mechanics, and function as an obstacle the player must navigate to avoid while moving at fast speeds. In our case, we have hills and trees surrounding the player, which both adds to the theme of enchanted forest as well as provides a clear, logical boundary to the track.

**You Mute, You Lose**

A design pattern that incorporates using specific sound effects/music that allows the player to be notified of incoming actions as well as their result. Utilizes sound effects to also allow players to be familiar with the pacing of that level.

**Teach Me How to Fight**

When players begin a game it is important to show them basic mechanics by introducing smaller less meaningful targets to practice on. In our case, our enemies are static and introduced in small packs to the player.

**Let's Take Things Slow**

A mechanics based pattern, this directly relates to our design problem as it describes how to slow down fast-paced action in games. Specifically, with difficult maneuvers or when players are first starting out, incorporating slowed-time movements or actions is helpful to grant players more time if their reaction times aren't quick enough yet. Incorporating affordances like a physical timer that allows players to hit a button within the time range, or bullet time when attempting fast-paced or high-precision movements will be helpful to aid newer players.

**I'm Doing it as Hard as I Can**

---

Gradually increasing the difficulty for a new player will help give them the time to become more adept at basic mechanics so that they may use them at higher levels later on. We used this for our turns (making the turns walled, then unwalled, then tighter turns), introducing our jumping on ramps mechanic and drifting/boosting mechanic in safer spaces earlier on in the track, and in the slow introduction to enemies in the track.

**This is About to Hurt**

Players can better prepare for incoming obstacles or battles when they are made aware of an upcoming engagement. When new players are learning a game, icons and modular assets can help them be made aware of what type of engagement is coming and how to properly react to that. One example in our game is our enemies' particles that are emitted and visible from a distance, potentially before the enemies themselves.

**Let's do it Again**

When beginning to tutorialize a player, giving them the ability to repeat without the fear of failure allows them to strengthen their understanding of what they are learning. Checkpoints also allow the player to repeat difficult portions of a level. Combined with an infinite live/restart system, players are able to go at their own pace before feeling accomplished.

**Chaos is the greatest form of Fun!**

Random elements in a competitive setting are usually not welcome, but their inclusion allows for less skilled players to be on a more equal playing ground, keeping their enjoyment alive.

**Sitting Through the Previews**

In a game that may have a lot of action going on that could be overwhelming for new players, having some kind of safe space or pause in gameplay that allows them to just observe the physical space they'll be dealing with can be helpful for them to plan their strategy ahead of time and know what to expect. In our case, we actually created a quick 6 or 7 second cutscene that had multiple cameras panning over different areas of the track before the player began.

Each of the above patterns was developed over the course of the semester. Some of the patterns were developed by this team specifically to address this design problem, but others were created by other teams earlier in the semester. All created patterns were linked using keywords and pattern relationships, allowing this team to discover a set of patterns that met their needs.

*2.8. Example of Pattern Use in an Industry Context*

Discussions with industry professionals have yielded the consistent opinion that fully pattern-driven development is impractical for a studio just beginning to use patterns. Thus, the adoption of such a development methodology would be at best a gradual process, and possibly not feasible given real-world production constraints. However, more limited use of pattern exercises and language to solve a specific design problem has proven successful in at least one case.

Developer Christopher Totten of Pie For Breakfast Studios has used these techniques to establish a level design paradigm for his successfully Kickstarted game Little Nemo and the Nightmare Fiends. His team was having trouble finding a shared frame of reference for their level design style. Totten conducted several pattern exercises, and the team produced several patterns that they were able to use to define, understand, and agree upon the paradigm they would use for level design within the game. One of the primary patterns is shown and linked and described in the below figure (Figure 1).

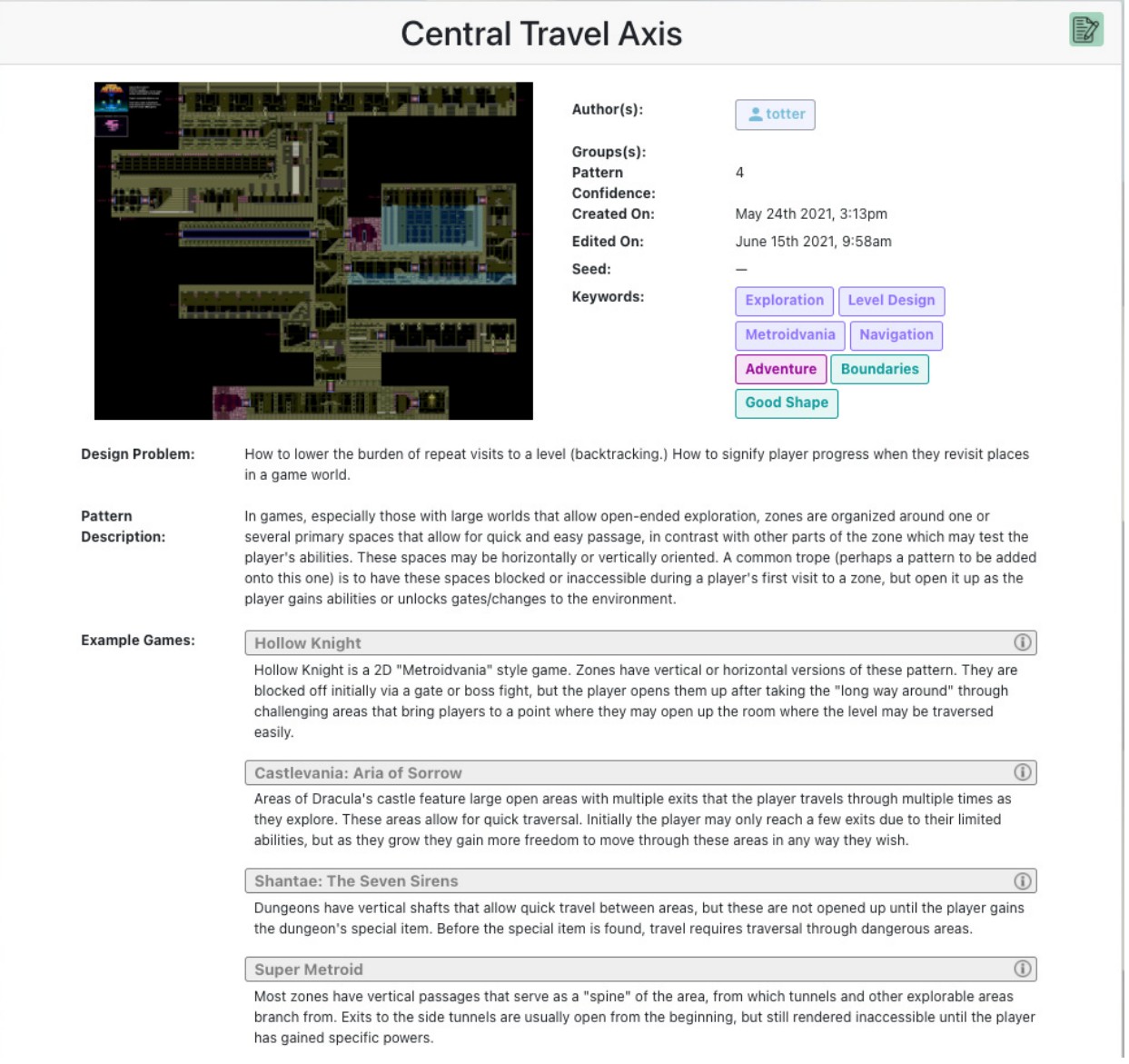

**Figure 1.** Example designer-created pattern.

### 3. Results

This project focused on the application of pattern theory in pedagogy and design practice rather than being structured as a formal study. Formal experimental results cannot be provided at this stage. However, a significant amount of data were generated in the form of completed design patterns and graded student projects. This growing dataset can be formally assessed in future research. Additionally, initial observations made during this project can be offered.

*3.1. Peer Review and Course Grading*

The results of the pattern exercises that were developed over the course of this project were assessed in several ways. The most direct was through grading by the instructor. Each pattern was considered in the context of existing design theory. In most cases, the students did not enter into the exercises with formal knowledge of this design theory. The consistency with which students articulated known design theory and with which the so-

phistication of that theory exceeded their previous understanding suggests that the exercise process was effective in helping students synthesize their experiential design knowledge and research into formal theory.

A second form of assessment was used in which students assessed each other's patterns and design projects. These assessments were then reviewed by the instructor. In assessing pattern exercises, the ability of the reviewing student to understand the theory being articulated by the students producing patterns was considered. In project assessment, the ability of the assessing students to identify the ways that patterns were implemented in the projects was considered. In most cases, the reviewing students' ability to perform both of these tasks and correctly identify and clearly articulate design techniques in their peer's work was higher than in previous courses that had not used patterns as part of the curriculum. This suggests that the patterns produced were an effective method of communicating design ideas between developers and in helping students integrate design ideas introduced in course texts.

Specifically for the course that produced the above example, students were assigned projects from other teams and asked to complete the following exercises:

1.  Look at the design document of your assigned group.

    a.  Compare it to what you saw in the demonstration.
    b.  In your readings and assignments document, list each pattern the group used.
    c.  Describe how well you think it was implemented and whether it had its intended effect.

2.  List as many techniques from Game Design and Architecture as you can find in their design. For each technique:

    a.  Is it part of a pattern, and which one?
    b.  Does the technique help solve the group's design problem? How?
    c.  What techniques do NOT help solve the design problem?
    d.  Do the techniques that do not relate to the design problem support techniques that do? If not, could they be removed?

This exercise produced two to three pages of analysis from each student. A few representative excerpts are included below. These are indicative of the level of understanding acquired by the students through the process of pattern derivation and use.

In response to exercise 1:

**1.**  Teach Me How to Fight (analyzed by Colin Yang)

I think the track/level itself serves as the "smaller target" for players to practice the game's mechanics on. The level includes all of the mechanics listed in the design and doesn't punish the player much if they fail. As such, I think the pattern serves its purpose well.

**2.**  Let's do it Again (analyzed by Colin Yang)

The design includes checkpoints that respawn the player at the beginning of the section for the mechanic that they failed. The track is also loopy, so that if the player fails, they land right back on the track, also making it easier to get right back into it. I think this pattern is implemented well.

**3.**  I'm Doing It As Hard As I Can (analyzed by Michael Iantosca)

This pattern is concerned with gradually increasing difficulty in order to compensate for a player's increase in skill level. This pattern actually applies to group 4's final level in a number of ways. Firstly, like I mentioned earlier, the more complex bits of the track come later on in a circuit, and scaling difficulty is inherent to any racing game, as the better the player becomes, the faster they will be able to get around the circuit.

Students accurately identified the ways that each pattern had been applied to the project games. When there were flaws in the implementation of the patterns, analyzing students identified them in a nuanced way and often suggested changes in implementation that would improve the effect of the pattern in the game.

In response to exercise 2:

1. Montessori Building Blocks (analyzed by Harrison Sims)

Pattern: This technique is connected to Teach Me How to Fight and Let's Do it Again because the technique is used to teach and then test, but over the course of the game. As the game goes on, I would expect the designers to teach the player new techniques that can be used to complete levels.

Analysis: This technique does help new players with fast and reaction gameplay because it is about teaching the player how to play the game basically, and the teaching will make the player better at the desired gameplay.

2. Access as a First Level Reward (analyzed by Harrison Sims)

Pattern: Architecture Boundaries and Let's Do it Again both apply to this technique, but for different reasons. Analysis: Architecture Boundaries because areas are "locked" until the player passes the current area using the new skill, like the jump or the power up loop. Let's Do it Again because in sections like the jump or the power up loop the player has multiple attempts to complete the task, and the reward is moving to the next section of the map. I think [the technique] does help with the design problem because it makes sure the player is able to accomplish the skills necessary for the fast and reaction-based movement of the later levels.

3. Negative space (analyzed by Michael Iantosca)

Pattern: This technique relates to the pattern It's Not All About Looks

Analysis: It helps define the boundaries of the play space. It helps to solve the design problem by creating a clear, continuous circuit for new player to traverse. The negative space created by the lack of high mountains and trees in the track immediately cue the player in on where they are supposed to be going and clearly define where the boundaries are.

4. Local symmetry (analyzed by Michael Iantosca)

Pattern: I'm Doing It As Hard As I Can.

Analysis: The track makes use of symmetries by repeating challenges from earlier in the track later on with increased challenge and complexity. In this way, it relates to the pattern I'm Doing It As Hard As I Can. It helps solve the design problem by allowing new players to try the challenges of the circuit multiple times in order to get better at them, even within a single run.

In the responses to this second exercise students show that they can connect the many design techniques that they have learned about in the class outside of the pattern process. These techniques are presented through the text for the class and through lecture and discussion. The ability to relate these techniques to the patterns that they have derived during the course grounds their evolving personal design frameworks with established design theory.

*3.2. General Observations across Student and Developer Projects*

A number of broad trends are visible across student developer patterns and projects. The initial patterns produced often state design principles that the developer is already familiar with. The type of pattern produced is easily influenced by the context in which the developer is working, whether it be a particular design project or the material being covered by a course. Although game designers need to be aware of this effect, it is not necessarily detrimental since it allows the developer to focus on the problem they are currently facing. Early attempts at completing pattern exercises are time consuming. Developers tend to gain speed with early repetition, but eventually slow in their process again as they learn to use the exercises to look for patterns relating to techniques that they do not already understand well. When using patterns in projects, groups of developers are able to make use of the patterns that they have become familiar with to create a shared vocabulary and communicate complex design concepts efficiently.

*3.3. Mapping the Teaching and Learning Functions to the Pattern Creation Process*

As part of the process of assessing the efficacy of using patterns as a pedagogical framework, it is useful to apply a widely understood set of pedagogical techniques, such as the learning and teaching functions proposed by Shuell and Moran [24]. The consideration of each of these twenty-two functions in respect to the specific steps in the creation and application of patterns as discussed here yields a potential mapping of their functions to this process. This mapping seems to be direct and robust enough to merit further formal analysis and suggests that this application of pattern theory could provide the basis for a viable pedagogy for the teaching of game design. The proposed mapping is shown in full below.

3.3.1. Preparation

- **Prior knowledge activation:** Designers are asked to choose seeds for the patterns they develop based on their prior knowledge, or in the case of students, concepts that have just been introduced through course texts.
- **Motivation:** Completion of the pattern exercise process is required by the developer's team mates as well as for the completion of game projects; both of these provide external motivation for developer persistence and contribution.
- **Expectations:** Exercises are introduced with full examples, and designers completing them identify uses of the pattern as part of the process of naturally identifying the utility of the process they are completing.
- **Attention:** In order to identify patterns and clarify their use in example games, designers must isolate the formal and functional techniques that those games contain which relate to their exercise.

3.3.2. Knowledge Manipulation

- **Encoding:** Developers create their own patterns, deciding how to express the ideas they have observed. They choose the pattern title and image to help solidify the concept the pattern captures.
- **Comparison:** During the first stages of the pattern process, designers compare the use of their seed technique across existing games. Later in the process, they identify the best applications of the pattern that they have articulated as it is applied in existing games and choose the most diverse applications as examples to include in their pattern.
- **Repetition:** Within a specific iteration of the process of pattern discovery, designers analyze ten or more games looking for uses of their seed technique. The larger process of language creation involves repeating the pattern creation process many times and reviewing the created patterns looking for connections.
- **Interpreting:** Designers must examine existing games which contain the techniques they are investigating, understand the use of those techniques, and then articulate the shared aspects of their purpose and implementation in the form of a pattern. Later, they must begin with a pattern and design a game that implements the pattern to achieve the previously stated purpose.
- **Exemplifying:** When completing an exercise, developers must provide examples of the use of the pattern. These examples usually differ from the games that were analyzed as the source for the pattern. Additionally, designers are encouraged to find the most diverse set of examples possible to illustrate the scope of their pattern.

3.3.3. Higher Order Relationships

- **Combination, integration, synthesis:** Individual patterns are created by observing and combining the purpose and implementation of techniques across games. Pattern languages are created by articulating the relationships between patterns in terms of subject, purpose and function.
- **Classifying:** Each pattern must be assigned a set of keywords to place it within the context of existing design theory. Three levels of keywords are provided: keywords

which identify the patterns' subject matter, categories which place it in an area of design, and properties which indicate its purpose.

- **Summarizing:** The description of the pattern is a summary of the analysis that the designer has undertaken to derive the pattern.
- **Analyzing:** Patterns are created through the analysis of a set of existing games; these must be decomposed and understood in terms of the seed technique of the pattern exercise.

### 3.3.4. Learner Regulation

- **Feedback:** As part of the language creation process, patterns are peer reviewed and revised to best form the connections necessary for the language.
- **Evaluation:** On project completion, projects are peer reviewed to analyze the efficacy of their implementation of the patterns.
- **Monitoring:** During the use of patterns in design projects, the implementing designers provide feedback to the designers that developed each pattern.
- **Planning:** The use of patterns in practical design projects is intrinsically a planning process wherein the designers use patterns to structure their design prior to implementation.

### 3.3.5. Productive Actions

- **Hypothesis generation:** The process of pattern formation consists of analyzing data and forming a hypothesis.
- **Inferring:** Designers take existing design knowledge, examine existing examples of its use, and infer the patterns that it forms.
- **Explaining:** Creating the textual artifact of a pattern using the provided template allows designers to articulate and explain the theory they have constructed. Patterns are then further used to explain the more complex composite concepts that form a complete game design.
- **Applying:** Using patterns as the basis of design in practical game projects allows designers to apply the concepts that they have articulated and validate their efficacy.
- **Producing and constructing:** From simple scene implementations using a signal pattern to complex full game designs, the practical execution of a design into a game provides designers with the opportunity to demonstrate their learning in functional game artifacts.

## 4. Discussion

### 4.1. A Rigorous Review Process and Future Metrics

As the basic process for creating patterns and applying them to design projects has solidified, it has become more feasible to design a rigorous set of metrics for measuring the accuracy and efficacy of patterns.

This process must include:

- a stable rubric for assessing patterns in terms of existing design theory;
- a persistent way to record the use of patterns in student projects;
- standardized assessment of playtest data for games using patterns;
- assessment of games developed by equivalent students without the use of patterns;
- a larger sample of industry developers producing and using patterns.

The rapid, iterative, and largely informal process used to develop this pedagogy has been remarkably productive, but in order to understand and validate the extent of its usefulness, more formal study is clearly required. The empirical study design used by Florez et al. in their Teaching Software Engineering Topics Through Pedagogical Game paper is applicable here with minor modifications. This study model will be applied in the Games@Northeastern: Studio Project discussed below.

*4.2. Limited Project Size*

One of the major limitations in the observed use of patterns to produce game projects in the classroom is the limited scope of the projects themselves. In each course, seven one-week projects are completed, followed by a single four-week project. The shorter projects produce a playable scene, and the four-week project produces a longer scene with more polish. These short projects often clearly show the function of a pattern, but they do not show its efficacy in the context of a complete game.

*4.3. The Benefits and Limitations of Student Developers*

The use of this process with students has produced higher quality analysis and clearer articulation of the design techniques used to produce projects. This effect may or may not be reflected in developers of greater experience. Every derived pattern provides new insight for students, and it is possible that experienced designers may derive less benefit from the process. To date, only three experienced designers have completed a significant number of pattern exercises and related their experience. All three have found the process useful and derived patterns that addressed their needs, but this number is far too low to make any generalization.

*4.4. Games@Northeastern: Studio Project*

During the summer of 2021, a new project began at Northeastern University to allow students to undertake large-scale development using patterns to guide design. In this project, up to fifty students will participate each semester for a year to produce a single complete game. This development process will be monitored from its inception to more rigorously evaluate the effect of patterns on its design. The outcomes of that project will be presented when they become available.

The design of this project is intended to leverage the use of patterns both to maximize student learning in the process and to allow all of the students in the studio to contribute to the design and have visibility into the high level and detailed design. The basic structure of the design process is as follows. Over the summer of 2021, students will all contribute design ideas and collectively select the most interesting subset. Selected ideas will be decomposed into their atomic mechanics and design techniques. Pattern exercises will be completed with these seeds. The resulting patterns will be connected into a pattern language. In five teams, students will examine the resulting language and select a subset of patterns that are densely connected to form the basis for a final design. One design will be selected, and all students will participate in the development of that game over the Fall and Spring semesters.

## 5. Conclusions

The idea of a pattern language was introduced by Alexander almost 45 years ago in a different field of study. Today, the idea of patterns is commonly understood in game design; however, the potential utility of a pattern language in game design education and practice has not been realized. Having identified the omission of design problems, presumption of functional completeness at the level of pattern languages, narrow topical focus, and lack of a concise, repeatable method for pattern production as shortcomings in existing implementations of patterns in game design, a more robust method for applying patterns to game design was created.

This new method addressed the above weaknesses through introducing a new pattern template, creating a defined process for connecting patterns into a language and assessing the scope of that language, creating a rubric for assessing pattern confidence and interconnectivity confidence, and generating an introductory set of pattern-building exercises. This method was iteratively applied in the classroom and by several industry designers allowing for its refinement and assessment.

This use of the pattern creation and application methods discussed in this paper allowed the students and developers involved to

- access the learning opportunities offered by the process of creating patterns;
- utilize the communication clarity created through a shared understanding of a pattern;
- practice thoughtful design through connecting practical design problems with the patterns that solve them.

On a small scale, the process has been successful in use with students and in limited use by developers in industry. However, more extensive rigorous study of the process and its results is needed and has begun.

**Supplementary Materials:** The following are available online at www.mdpi.com/article/10.3390/info12100393/s1.

**Funding:** This research received no external funding.

**Acknowledgments:** The analysis of the methods discussed in this paper would not have been possible without the dedicated course work of the students in the GAME and GSND programs at Northeastern University. In particular, this paper cites example patterns and analysis from the following students as representative of the larger body of analyzed work: Simon Astor, Izzy Conner, Liam Cristello, Meredith Goujon, Magen Hocker, Becca Malcolm, Michael Iantosca, Leylanah Mitchell, Danny Son, Jack Speake, Harrison Sims, Liam Sean Thornton, and Colin Yang. Additionally, the pattern work of the industry designers Christopher Totten and Link Hughes is appreciated.

**Conflicts of Interest:** The author declares no conflict of interest. The founding sponsors had no role in the design of the study; in the collection, analyses, or interpretation of data; in the writing of the manuscript; and in the decision to publish the results.

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
