# Peer review of "Application of Pattern Language for Game Design in Pedagogy and Design Practice"

_information, doi:10.3390/info12100393_

Round 1
Reviewer 1 Report
I think it is unfair to assert, as you do on line 119, that Joris Dormans and I considered the design pattern library in Game Mechanics: Advanced Game Design to be "functionally complete." We did not make that assertion. We were well aware that more patterns would emerge.
Author Response
You are completely right. You clearly state that you do not consider your language complete on page 165 of Game Mechanics: Advanced Game Design. I apologize for the confusion, and I will revise that section of the paper immediately. I think I may have conflated a section of the 'gang of four' book with yours as I was re-reading it at the time I read Advanced Game Design. Also, thank you for your work on design patterns, your mechanics-related patterns are by far the most sophisticated and well-conceived in the field and I hope that my work will help new designers to develop patterns of that quality in other areas of design.
Reviewer 2 Report
In the submission entitled „Application of Pattern Language for Game Design in Pedagogy and Design Practice“ aspects of game design and particularly implementations of design patterns are discussed. The manuscripts starts with an overview on current problems and methodological weaknesses of pattern impklementations and finally leads to a proposed method to overcome existing problems.
The manuscript is clearly structured and written in a comprehensible way. The topic does not exactly fall within my expertise. Thus, I cannot decide on some oft he contentual aspects oft he text. Basically, from my point of view, it provides a useful and novel insight into the developing field of game design. Thus, I think it shouzld be of interest fort he readership of Information.

Author Response
Thank you for your review. I have made minor changes to clarify the intent of the research design and to highlight the ongoing validation of the methods described in this article. Additionally, I clarified the utility of the research outside of an educational setting.
Reviewer 3 Report
This paper describes the creation of a pattern template that can be used to define design patterns and create a pattern language for design issues.
This methodology was used to teach design concepts to students from games studies programs.
My main concerns about the paper are:
1 - What is the innovation introduced by this article if the process in question was published in the book (Pattern Language for Game Design) and has a website to support its use? It was the use of the methodology in teaching students from game studies?
2 - The proposed methodology allows the creation of a Pattern Language to solve a specific problem of game design, which is useful for the practice and learning of game design concepts.
But is the methodology adequate for the game design of a game as a whole? How?
3 - As said by the author, the language created consisted of 80 to 117 patterns and are not considered to be functionally complete, or even sufficient to fully serve as a basis for pattern driven game-development. So, the main objective is serving only as a learning tool for students?
4 - The supplementary material is not necessary in the paper because one example is partially presented. Besides, the author can always mention examples of patterns created, which are available on the website of the project.
5 - The methodology was not scientifically validated yet, the results presented are mainly based on the empirical observation and opinions of students and teachers. Even because if the novelty of the article is about the use of the methodology in teaching game design, their validation is crucial. Scientific validation is needed.
Some mistakes in the text:
- The references at the end of the phrases appear for example "... ... scope of game design. [10][22]" and must be "... scope of game design [10][22]."
- In the introduction section appear:
"... Four Keys to Fun [19]," and must be "Four Keys to Fun [20],"
"Pattern Language for Game Design. [4]" and must be "Pattern Language for Game Design [3]."
“that of Gamma et. al. ” and must be “that of Gamma et. al. [16]”
“papers Game Design Techniques for 150 Software Engineering Management Education [27] ” and must be “papers Game Design Techniques for 150 Software Engineering Management Education [??] ” where ?? is the following reference:
P. Letra, A. C. R. Paiva and N. Flores, "Game Design Techniques for Software Engineering Management Education,"
2015 IEEE 18th International Conference on Computational Science and Engineering, 2015, pp. 192-199, doi: 10.1109/CSE.2015.42.
- In section 2 appear:
“As stated in the section An alternative application“ and must be “As stated in the section an alternative application”
“Author(s): All of the designers who contributed to the creation of the problem.” and “Author(s):” must be in bold.
“here in Figures x, x, and x” and must include the figures numbers
“experienced designers. M importantly they” and must be “experienced designers. Most importantly they”
Author Response
Thank you very much for the useful and specific feedback. I have addressed the formatting issues that you pointed out. I have also made the following changes to address each of your five concerns with the content of the paper.
This paragraph was added to the end of the Introduction:
"The work described in this paper differs from and extends the work published in the book Pattern Language for Game Design [3] in that it proposes a formal ontology for the pattern template and describes the ways that the pattern creation process maps to the learning and teaching functions described by Shuell and Moran [28]. Additionally, this paper describes the application of the pattern creation process in both an educational setting and as part of practical development in an industry setting. This establishes methods for applying a clear pattern language creation process that, through iterative development, have reached a stable state where formal validation can begin. The validation process is described in the Discussion section of this paper and is ongoing. Results will be presented in future publications as they become available."
I believe this clarifies the intent of the paper to address concerns 1 and 5. Concern 5 is also addressed in the first paragraph of the Results section where the scope of the research is stated. It is also stated that data for formal analysis is being gathered. Additionally, the beginning of the Discussion section of the paper describes the ongoing formal research that is being conducted as part of a specific project.
Concerns 2 and 3 are addressed by the following addition to the end of the section Proposal for individual patterns and languages:
"Finally, the process allows individual developers to organize their personal design knowledge as best fits their needs. Thus, while individual patterns and small problem-driven languages are created independently, they are interoperable and can be combined to address the problem-set that defines the entirety of a given design."
In regards to the supplementary material, it is my understanding that it will not be presented in the body of the artilcle. I felt that it should be available as it is not material available on the Pattern Library website. All specific patterns are available on the website, however, the supplementary material includes the student work for the exercise that produced the listed pattern. This work is not captured on the website or reproduced in the preceding book. The other supplementary material is the detailed ontology. I included a reduced view of the ontology in the body of the article but am including the full version in the supplementary materials as it is in the formal syntax that would allow other researchers to import it into any OWL2 ontology editing tool for their use. All of that said I am willing to remove these materials at the publisher's request if they are not desired.
I hope these changes adequately address your concerns and am of course willing to make further edits to the article if needed.
Thank you again for your time.
Round 2
Reviewer 3 Report
The authors clarify in the text my main concerns about the paper, thus I can accept it now for publication.
Please check the year of the references 17 because it is not correct.
Author Response
I have corrected the citation. That was a good catch! The conference that the paper was presented at was held in that location from 2001 to 2004, but the paper was presented in 2004. The correct citation I have used is:
17. Hunicke, Robin, Marc LeBlanc, and Robert Zubek. "MDA: A formal approach to game design and game research." Proceedings of the AAAI Workshop on Challenges in Game AI. Vol. 4. No. 1. 2004.